# Non-canonical functions of a mutant TSC2 protein in mitotic division

**Mary-Bronwen L. Chalkley**[1], **Rachel B. Mersfelder**[1], **Maria Sundberg**[2], **Laura C. Armstrong**[3], **Mustafa Sahin**[2], **Rebecca A. Ihrie**[1,4]*, **Kevin C. Ess**[1,3]*

**1** Department of Cell & Developmental Biology, School of Medicine Basic Sciences, Vanderbilt University, Nashville, Tennessee, United States of America, **2** Rosamund Stone Zander Translational Neuroscience Center, Department of Neurology, Harvard Medical School, Boston, Massachusetts, United States of America, **3** Department of Neurological Surgery, Vanderbilt University Medical Center, Nashville, Tennessee, United States of America, **4** Department of Pediatrics, Vanderbilt University Medical Center, Nashville, Tennessee, United States of America

* rebecca.ihrie@vanderbilt.edu (RAI); kevin.ess@vumc.org (KCE)

## Abstract

Tuberous Sclerosis Complex (TSC) is a debilitating developmental disorder characterized by a variety of clinical manifestations. TSC is caused by mutations in the *TSC1* or *TSC2* genes, which encode the hamartin/tuberin proteins respectively. These proteins function as a heterodimer that negatively regulates the mechanistic Target of Rapamycin Complex 1 (mTORC1). TSC research has focused on the effects of mTORC1, a critical signaling hub, on regulation of diverse cell processes including metabolism, cell growth, translation, and neurogenesis. However, non-canonical functions of TSC2 are not well studied, and the potential disease-relevant biological mechanisms of mutations affecting these functions are not well understood. We observed aberrant multipolar mitotic division, a novel phenotype, in *TSC2* mutant iPSCs. The multipolar phenotype is not meaningfully affected by treatment with the inhibitor rapamycin. We further observed dominant negative activity of the mutant form of TSC2 in producing the multipolar division phenotype. These data expand the knowledge of TSC2 function and pathophysiology which will be highly relevant to future treatments for patients with TSC.

## Introduction

Tuberous Sclerosis Complex is a genetic disorder with a broad spectrum of phenotypes, affecting virtually every organ system [1]. This disorder occurs in approximately 1 in 6,000 people and is marked by mutations in *TSC1* or *TSC2* which code for hamartin and tuberin, respectively [2–5]. Loss of function mutations of either *TSC1* or *TSC2* are sufficient to cause pathogenesis, though mutations in *TSC2* have been associated with more severe phenotypes [3]. An autosomal dominant inherence pattern is possible, but *de novo* mutations are much more common, and are seen in the majority of patients with TSC [6]. Current models of TSC pathogenesis suggest a possible two-hit hypothesis wherein a loss-of function mutation is required in both copies of either *TSC1* or *TSC2* in order to produce disease [7]. However, not all hamartomas found in patients with TSC reliably demonstrate loss of heterozygosity, suggesting that haploinsufficiency, post-translational inactivation of remaining protein, or dominant negative

**Funding:** Yes. This research was supported by awards R01NS118580 (RAI, KCE), R01NS118580-

S1 (MBLC), and T32 HD007502 (MBLC). The funders had no role in study design, data collection and analysis, decision to publish, or preparation of the manuscript.

**Competing interests:** MS reports grant support from Novartis, Biogen, Astellas, Aeovian, Bridgebio, and Aucta. He has served on Scientific Advisory Boards for Novartis, Roche, Regenxbio, SpringWorks Therapeutics, Jaguar Therapeutics and Alkermes. The other authors have no competing interests to declare. This does not alter our adherence to PLOS ONE policies on sharing data and materials.

activities of mutant TSC1 (hamartin) or TSC2 (tuberin) may also underlie some disease phenotypes [8–10].

Hamartin and tuberin are canonical negative regulators of the essential mammalian/mechanistic Target of Rapamycin (mTOR) pathway [1, 11]. Hamartin and tuberin are constitutive inhibitors of RHEB GTPase (Ras homolog enriched in the brain), an upstream regulator of mTOR complex 1 (mTORC1) [12–14]. The principal identified effect of *TSC1* and *TSC2* loss of function mutations is thus unchecked overactivity of mTORC1 [12]. mTORC1 promotes protein synthesis, cell growth, and proliferation through phosphorylation of downstream targets, classically including the activation of S6 kinase (S6K) and the inhibition of eukaryotic initiation factor 4E-binding proteins (4E-BP) [12, 15–20].

A large proportion of patients with TSC present with epilepsy and are found to have extensive cortical hamartomas (also known as cortical tubers) of the brain in which lamination is disordered and dysmorphic neurons are present [1]. A diagnostic characteristic of cortical tubers is the presence of "giant cells", so named due to their large size. Giant cells are thought to have abnormal differentiation and can exhibit both astrocytic and neuronal features. Though the cell of origin of giant cells remains controversial, it has been repeatedly observed that giant cells can be multinucleated, a phenotype that can arise from multipolar division [21–24].

During cell culture experiments with induced pluripotent stem cells (iPSC) derived from a patient with TSC, we observed a high incidence of multipolar dividing cells. This suggests a possible mechanism that may underlie the multinucleated giant cells found in cortical tubers. In this report, we present evidence that a TSC2 loss of function mutant, containing a 6 amino acid deletion within the C-terminus, exhibits a dominant negative function leading to multipolar spindle formation. This finding highlights a possible contributing factor to the disruptions of normal neuronal migration and cortical development seen in patients with TSC.

## Materials and methods

Use of human stem cells. Deidentified cells from human subjects were used as previously described (Sundberg 2018; Armstrong 2017). These initial publications had details of derivation including informed consent and institutional approval. For line 77 (Sundberg et al. 2018), subject was recruited through Boston Children's Hospital and protocol approved by Boston Children's Hospital (Boston, USA) IRB (P00008224). Written informed consents were obtained from all participants and/or their parents as appropriate.

### Resource availability

**Corresponding author.** Further information and requests for resources and reagents should be directed to and will be fulfilled by the co-corresponding author, Kevin Ess (kevin.ess@vumc.org).

**Materials availability.** This study did not generate new unique reagents.

### hiPSC Cell Culture

iPSCs were grown as colonies on Matrigel (Corning, Corning, NY) coated 6 well plates or glass bottom 35mm plates (Cellvis, Mountain View, CA) in mTeSR1 medium (StemCell Tech, Vancouver, Canada), replaced daily, maintained at 37˚C and 5% $CO_2$, and passaged as needed with ReLSR (StemCell Tech).

## Cell treatment

Cells were grown as described above. When cells reached 50% confluence, they were treated with 0.4 nM or 30 nM rapamycin (Tocris Bioscience, Bristol, United Kingdom) or vehicle DMSO (Sigma Aldrich, St. Louis, MO) for 24 or 48 hours. Rapamycin was prepared in DMSO and added directly to the media. Cells were then lysed or fixed as described below.

## Whole cell extracts

Cells were washed with 1X PBS with 10 μM sodium vanadate. Cells were lysed using lysis buffer (1% Triton X-100 in STE [100mM NaCl, 1mM EDTA, 10 mM Tris pH 8.0], PhosSTOP (Sigma Aldrich), PIC (Sigma Aldrich), 50 μM MG132 (Sigma Aldrich), 1 μM PMSF (Sigma Aldrich)) added directly to the plate followed by scraping of the cells. Lysate was then sonicated for 3 seconds on power 3 at 4˚C, followed by centrifugation for 15 minutes at 16,000 x $g$ at 4˚C. Supernatant was kept at -80˚C until further analysis.

## Immunoblotting

Samples were prepared by mixing whole cell extracts with 4X Laemmli Sample Buffer (Bio-Rad, Hercules, CA) and then boiling for 5 minutes at 105˚C. Gel electrophoresis was performed on a 4–12% Bis-Tris gel (ThermoFisher, Waltham, MA) with 1X NuPage MOPS running buffer (ThermoFisher) at a constant voltage of 140V for 90 minutes. Proteins were transferred to a PVDF membrane (ThermoFisher) in transfer buffer on ice in the cold room (4˚C) overnight at a constant current of 33 mA. Post-transfer, the membrane was stained with Ponceau S solution (Sigma Aldrich) for 5 minutes to determine total protein loaded into gel lanes. Membranes were washed 3X in ddH2O. Membranes were blocked in 5% non-fat milk (RPI Corp, Mount Prospect, IL) in Tris-buffered saline (Corning) with 0.1% Tween 20 (Sigma Aldrich)(subsequently termed TBS-t) for 1 hour at room temperature with agitation. All primary and secondary antibodies were diluted in TBS-t with 5% non-fat milk, and the membrane was washed 3X with 0.1% TBS-t afterwards. Antibodies are listed in Table 1. Primary antibodies were incubated overnight at 4˚C with agitation. Secondary antibodies were incubated for 1 hour at room temperature with agitation. The membrane was imaged using ECL developing reagents (ThermoScientific) and a CCD Imager—AI600 (General Electric, Boston, MA).

## Immunofluorescence

All cells were fixed by incubation in 100% methanol for 10 minutes at -20˚C. Fixed samples were blocked with blocking buffer [PBS (Corning), 1% Normal Donkey Serum

**Table 1. Antibodies used in these experiments.**

| Antibody | Manufacturer | Catalog | Dilution | Application |
|---|---|---|---|---|
| Alpha Tubulin (TUBA) | Invitrogen | MA1-80017 | 1:500 | Immunostaining |
| GAPDH | Cell Signaling Technologies | 2118 | 1:2000 | Western Blot |
| TUBERIN (TSC2) | Cell Signaling Technologies | 4308 | 1:2000 | Western Blot |
| phospho-S6 Ribosomal Protein (Ser 240/244) | Cell Signaling Technologies | 5364 | 1:1000 | Western Blot |
| S6 Ribosomal Protein | Cell Signaling Technologies | 2217 | 1:1000 | Western Blot |
| Goat anti-Rat IgG, Alexa Fluor 594 | Invitrogen | A-11007 | 1:1000 | Immunostaining |
| Goat anti-Rat IgG, Alexa Fluor 568 | Invitrogen | A-11077 | 1:1000 | Immunostaining |
| Goat anti-rabbit IgG, HRP | Invitrogen | 65–6120 | 1:5000 | Western Blot |

(ThermoScientific), 1% BSA (Sigma Aldrich), 0.1% Triton X-100] for 1 hour at room temperature. Primary antibodies were diluted in blocking buffer and then incubated overnight at 4˚C. Secondary antibodies were diluted in blocking buffer and then incubated 1 hour at room temperature in the dark. Antibodies are listed in Table 1. Images were acquired using a Prime 95B camera mounted on a Nikon spinning disk microscope using a Plan Apo Lambda 20x objective lens. 9 randomized images per genotype and condition were captured. The software used for image acquisition and reconstruction were NIS-Elements Viewer (Nikon, Tokyo, Japan) and ImageJ (FIJI).

## Image analysis

Images were blinded prior to scoring. CellProfiler was then used to quantify the total number of cells in each image. Subsequently, cells were manually identified to be in mitosis and in prophase, metaphase, anaphase, or telophase. The number of cells with more than two mitotic spindles was also recorded. Two-way ANOVA and student t-tests were run using Prism (GraphPad, San Diego, CA) on the collected data.

## DNA content flow cytometry

iPSCs were grown to 70% confluency as described above. Cells were washed with 1X PBS before incubating with ReLSR (StemCell Tech) for 30 seconds at room temperature. ReLSR was aspirated and the cells were allowed to further incubate at room temperature for 3 minutes. Cells were resuspended in 1X PBS and spun down 300 x g for 5 minutes. Supernatant was aspirated and cells were resuspended in 300μL 1X PBS. Cells were vortexed to remove clumps. 700μL ice-cold 100% Ethanol (Decon Laboratories, King of Prussia, PA) was added to the cells and the sample was incubated at -20˚C for at least 10 minutes up to overnight. Cells were counted using Trypan Blue (Corning) on a Countess II (Invitrogen). Cells were washed twice with 1X PBS by spinning down at 1000 rpm for 5 minutes followed by resuspension in staining mix [10mL 0.1% Triton X-100 (ThermoScientific), 10μL RNAseA (100 mg/mL)(Qiagen, Hilden, Germany), 10μL DAPI (1mg/mL)(ThermoScientific)]. Cells were resuspended to a uniform final concentration ($1 \times 10^6$ cells/mL staining mix). Cells and staining mix were passed through a cell-strainer cap on FACS tubes (BD Falcon). Cells were incubated overnight at 4˚C. Stained cells were then run slowly on a 3-Laser LSRII flow cytometer (<100 events/second). Stained cells were expertly gated for live, single cells. Data was stored and processed on Cytobank. Events were recorded as G1 phase for the first DAPI peak (2n) and G2/M phase for the second DAPI peak (4n) of the DAPI histogram. Events between the peaks were recorded as S phase.

## siRNA transfection of hiPSCs

Cells were grown as described previously. When iPSCs reached 50% confluency, 1 μM siRNA (Horizon Accell SMARTpool, Waterbeach, United Kingdom) in DMEM/F12 (Gibco, ThermoScientific) was added to the cells and left on for 24 hours. Media was changed to mTeSR1 after 24 hours and cells were allowed to grow normally for 48 more hours. iPSCs were then fixed or lysed.

## Results

### Patient-derived *TSC2* mutant iPSCs display multipolar division

We have been using an allelic series of isogenic iPSCs derived from a patient with a C terminal 6 amino acid in-frame microdeletion (c.5238_5255del, p. His1746_Arg1751del) in exon 41 of

*TSC2* [25]. An accompanying set of isogenic cells was generated from the heterozygous patient-derived iPSC (TSC2 +/LOF); in one line the second, wild type allele was also mutated (TSC2 LOF/LOF) and in the other the heterozygous mutant allele was corrected to wild type (WT) as previously described [25]. A mutant TSC2 protein is produced from the allele harboring the 6 aa in frame microdeletion found in the TSC2 +/LOF and TSC2 LOF/LOF cells, although the expression level of this mutant protein is decreased relative to TSC2 levels seen in other wild-type iPSCs and the corrected patient-matched line (Fig 1a and 1b).

During standard cell culture maintenance, we observed that patient-derived TSC2 +/LOF iPSC cells exhibited a higher incidence of multipolar division, with cells including three or more mitotic poles as opposed to the typical two expected during mitosis (S1a Fig). This phenotype was detected in both the original patient-derived line as well as the homozygous LOF line (TSC2 +/LOF and TSC2 LOF/LOF) (Fig 1d). To determine the rate of multipolar division in the mutant patient iPSC line, we counted the total nuclei present and then scored the total number of nuclei undergoing division within this population. The average rate of division was not significantly different in the wild type compared to mutant iPSCs (TSC2 +/LOF and TSC2 LOF/LOF) (S1b Fig). Further, no significant difference was found between genotypes for the proportion of cells in each phase of mitosis nor the cell cycle (S1c–S1g Fig). Nuclei undergoing division and exhibiting more than two spindle poles were considered to be multipolar, and the total number of such nuclei was counted. A significant increase in multipolar nuclei as a percentage of dividing nuclei was observed in TSC2 LOF/LOF iPSCs compared to matched wild

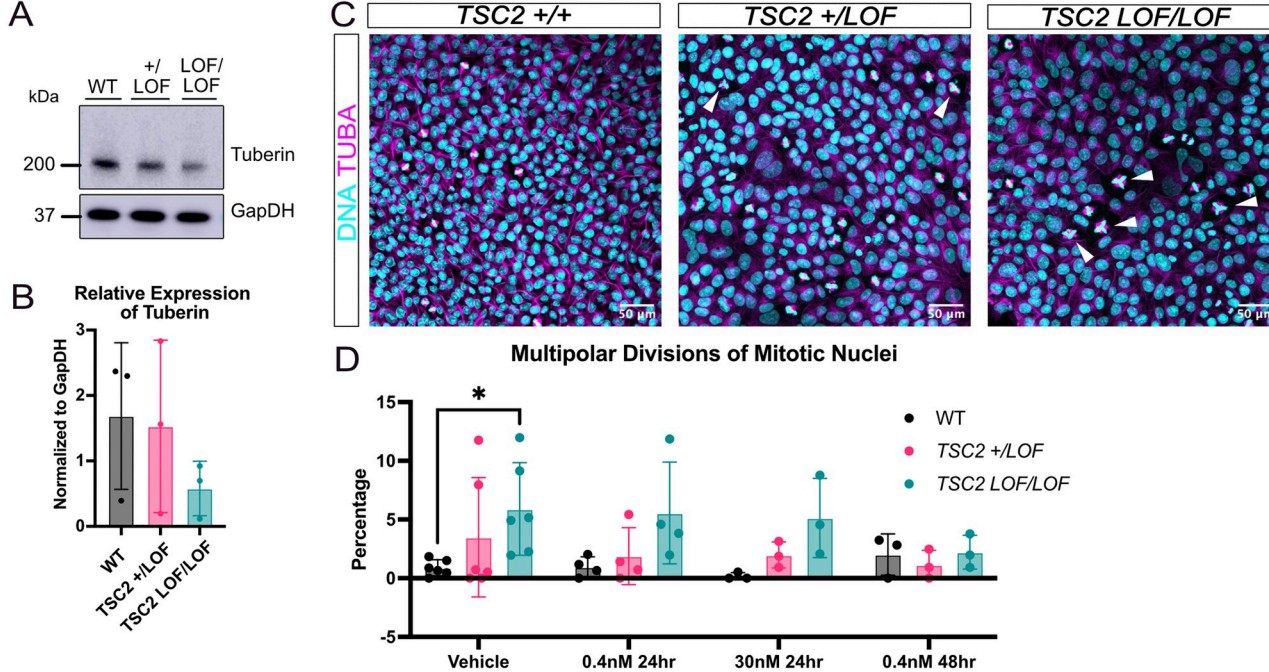

**Fig 1. Patient-derived TSC2 loss of function mutant induced pluripotent stem cell cultures have multipolar mitotic cells.** (a) Representative immunoblot showing protein expression of tuberin in patient iPSCs. (b) Quantification of mean relative expression of tuberin in immunoblot shown in A (across 3 technical replicates using these lines) Error bars = SD. (c) Average proportion of multipolar dividing nuclei by genotype. Vehicle n = 6; 0.4nM rapamycin for 24 hours n = 4; 30nM rapamycin for 24 hours n = 3; 0.4nM rapamycin for 48 hours n = 3 independent replicates per genotype. Genotype p = 0.0013. * refers to p = 0.0471 [Two way ANOVA and Dunnett's multiple comparisons test]. Error bars = SD. (d) Representative immunofluorescence images of induced pluripotent stem cells using TUBA (magenta) to identify mitotic cells and Hoechst (teal) to identify DNA. White arrows indicate multipolar dividing nuclei. Scale bars = 50 μm. (e) Representative immunoblot showing protein expression of vehicle and rapamycin treated cells in c.

type iPSCs (p = 0.0471) (Fig 1c). TSC2 +/LOF iPSCs showed an increase in multipolar nuclei compared to wildtype (Fig 1c). To determine if the multipolar phenotype was due to increased mTORC1 activity, each iPSC line was treated with rapamycin, a potent mTORC1 inhibitor, at different concentrations and for different durations (0.4 nM rapamycin for 24 hours, 30 nM rapamycin for 24 hours, 0.4 nM rapamycin for 48 hours). No significant differences in the percentage of mitotic cells were observed in vehicle treated vs all rapamycin dose and duration treated iPSCs (S1b Fig). Interestingly there was no significant change in number of multipolar nuclei with different rapamycin dose and duration of treated TSC2 mutant iPSCs (TSC2 +/LOF and TSC2 LOF/LOF) (Fig 1c) though at longest exposure there was a trend towards decreased multi-polarity in all genotypes. Further, we observed no significant change in percentage of mitotic cells in each phase of mitosis for any genotype when treated with rapamycin (S1c–S1f Fig).

## Multipolar division phenotype is not observed in TSC2 knock out iPSCs

We sought to replicate these observations in another iPSC line. To test the impact of the presence or absence of tuberin, we used a TSC2 CRISPR-engineered knock out iPSC line that did not originate from a patient and produces no tuberin protein, originally described in [26] (Fig 2a and 2b). When repeating the approach shown in Fig 1 with this line, few to no multipolar dividing nuclei were observed in the TSC2 knock out iPSCs (Fig 2c and 2d). Moreover, the average percentage of dividing nuclei were not significantly different between the isogenic wild type and TSC2 knock out iPSCs (S2a Fig). No significant differences were found between

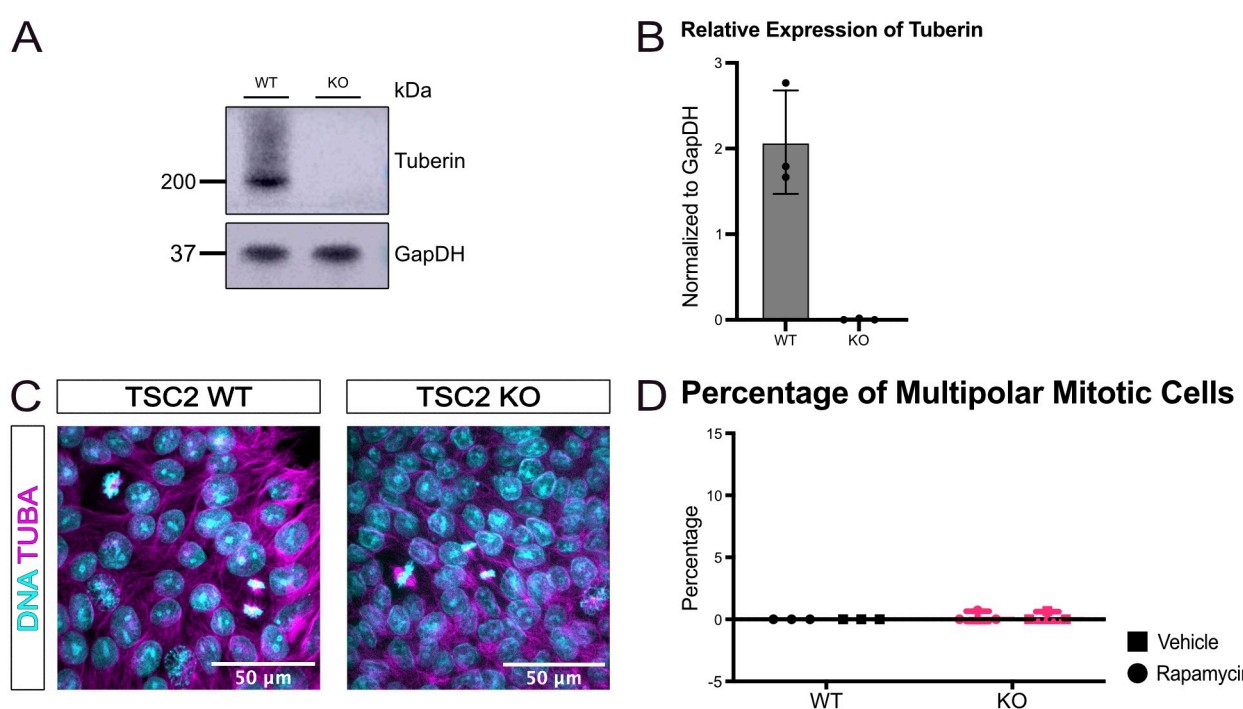

**Fig 2. TSC2 knock out induced pluripotent stem cell cultures have no multipolar mitotic cells.** (A) Representative immunoblot showing tuberin expression in TSC2 knock out iPSCs. (B) Quantification of relative expression of tuberin (across 3 technical replicates using these lines). (C) Representative immunofluorescence images of induced pluripotent stem cells showing expression of TUBA (magenta) and Hoechst (teal) to identify DNA to identify mitotic cells. Scale bars = 50 μm (D) Quantification showing average percentage of mitotic cells that have the multipolar division by genotype. N = 3 independent replicates per genotype. No significance [paired students t test]. Error bars = SD.

wild type and TSC2 knock out iPSCs when examining the proportion of cells in each phase of mitosis or cell cycle stage (S2b–S2f Fig).

## Patient-derived mutant TSC2 exhibits a dominant negative effect

Since the multipolar division phenotype was observed in the iPSCs with the mutant tuberin but not the *TSC2* KO iPSCs, we next tested whether the mutant version of TSC2 that is expressed in patient-derived iPSCs exerts a dominant negative effect. We performed a transient siRNA knock down in the patient mutant lines to remove the mutant TSC2 protein. Loss of TSC2 protein was observed in all iPSCs treated with siRNA against *TSC2* (Fig 3a). The proportion of multipolar division significantly decreased between the scramble control and siTSC2 treated patient TSC2 +/LOF iPSCs (p = 0.0051) (Fig 3c). The percentage of cells in mitosis increased in the *TSC2 +/LOF* siTSC2 treated iPSC compared to scramble control cells (Fig 3b). Interestingly, the average rate of mitosis and proportion of multipolar divisions did not significantly differ between scramble control and siTSC2-treated cells in the patient wild type. The same was true in scramble control and siTSC2-treated *TSC2* LOF/LOF iPSCs (Fig 3b and 3c).

## Discussion

TSC is a prototypical neurogenetic disorder whose study has led to many insights into the normal process of brain development as well as the pathogenesis of epilepsy, autism, and intellectual disabilities. While the disorder is caused by mutations in *TSC1* or *TSC2* genes, patients with *TSC2* mutations typically have more severe symptoms [1, 3]. In this study, we chose to

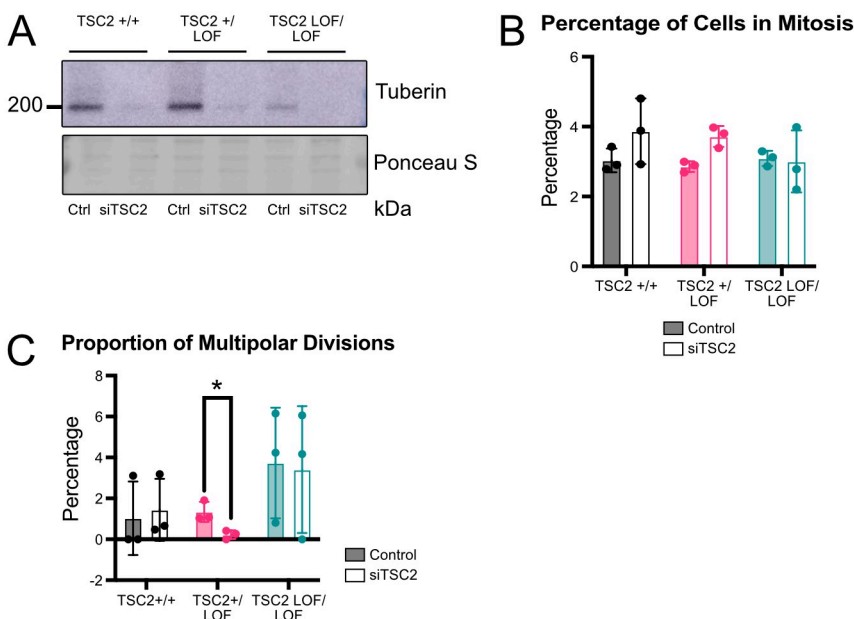

**Fig 3. siTSC2 induced pluripotent stem cells have multipolar mitotic cells.** (A) Representative immunoblot of induced pluripotent stem cells showing expression loss of Tuberin in siTSC2 iPSCs. (B) Quantification showing average percentage of cells in mitosis per genotype and treatment. N = 3 independent replicates per genotype and treatment [Two way ANOVA]. Error bars = SD. (C) Quantification showing average percentage of mitotic cells that have the multipolar division per genotype and treatment. N = 3 independent replicates per genotype and treatment, Genotype x treatment p = 0.0060, treatment p = 0.0283, * refers to p = 0.0051 [Two way ANOVA with Šidák's multiple comparisons test]. Error bars = SD.

use an isogenic allelic series of iPSCs from a patient with a *TSC2* mutation [25]. This particular *TSC2* mutation (c.5238_5255del, p. (His1746_Arg1751del) in exon 41) has been frequently reported. In fact, in multiple studies profiling mutations in patients with TSC throughout the world, this same 6 aa deletion was found to be one of the most common mutations [6, 27–30]. The impact from this specific mutant allele thus has important consequences for many patients with TSC globally.

Approximately 90% of TSC patients experience epilepsy which is thought to originate from cortical tubers [31]. A diagnostic characteristic of cortical tubers is the presence of giant cells. Prior studies of these lesions include reports that a subpopulation of cells are multinucleated [21–24]. Multinucleated cells can arise from several different mechanisms. One such mechanism is multipolar division, in which a cell has more than two mitotic spindle poles [32]. A cell undergoing multipolar division may then not complete cytokinesis, resulting in a single cell with more than one nucleus [33]. As multinucleated giant cells are located within cortical tubers, an unanswered question is whether cell division abnormalities are involved in disruption of early brain development and the formation of cortical tubers in patients with TSC. We found that iPSC cultures with both heterozygous and homozygous *TSC2* mutation include a subpopulation of cells undergoing multipolar division, indicating a role for tuberin in proper division during mitosis within this cell type. A possible contributing mechanism to the formation of multinucleated giant cells in cortical tubers from mutant tuberin may thus be abnormal mitosis with failed cytokinesis through multipolar division. Further study will be required to determine such a mechanism.

When we investigated *TSC2* KO iPSCs, which lack all tuberin protein, we did not find any nuclei undergoing multipolar division. As the complete absence of tuberin did not result in multipolar cells, expression of a mutant tuberin protein lacking 6aa seems to be required to trigger multipolar division. To test this possibility, we knocked down tuberin with siRNA and saw that in the TSC2 heterozygous iPSCs, upon treatment of siTSC2, there was an increase in the percentage of cells in mitosis and a decreased incidence of multipolar division. These findings indicate a dominant negative effect of the mutant tuberin against the wild type tuberin. The patient wild type and *TSC2 LOF/LOF* iPSCs did not show the dominant negative effect likely due to neither genotype having both a wild type copy and a mutant copy of tuberin. Dominant negative activity from a heterozygous TSC2 mutation, as indicated by these data, could indicate a novel mechanism of disease that has not been widely considered for TSC. Future studies are also needed to determine if any other *TSC2* mutations similarly lead to an increase in multipolar dividing cells.

This multipolar phenotype is potentially ameliorated by rapamycin, as indicated by the lower multipolar division rate seen in each genotype after the longer 48 hour exposure. Previous studies on tuberin function have primarily focused on mTORC1-dependent phenotypes, but more investigation into mTORC1-independent tuberin functions is strongly needed as some aspects of the TSC phenotype appear to be resistant to treatment with rapamycin or structurally related drugs [34–38].

The TSC heterodimer is composed of hamartin and tuberin [2, 14]. Hamartin has been shown to localize to the centrosome. Further, hamartin and hence the entire TSC heterodimer has been shown to interact with PLK1 (polo-like kinase 1), a kinase involved in the duplication of centrioles [39, 40]. Additionally, the transforming acidic coiled-coil 3 protein (TACC3) was found to be necessary to localize phosphorylated tuberin (p-TSC2 Ser 939) to the spindle pole, and TACC3- and TSC2-deficient cells were determined to have an increase in binucleated cells [41–43]. These data, combined with the findings from this study, indicate a role for tuberin and the TSC complex in centriole duplication, suggesting a possible mechanism that may underlie multipolar division.

## Conclusions

In summary, we determined that the patient mutant copy of TSC2 promotes aberrant multipolar division that is not significantly changed by treatment with the mTORC1 inhibitor rapamycin (Fig 1, S1 Fig). Complete loss of TSC2 does not produce multipolar division (Fig 2). Knockdown of mutant TSC2 shows a possible dominant negative effect with regards to this multipolar division (Fig 3). We anticipate that the results presented here will be a catalyst for further analysis of additional TSC patient-derived iPSCs for the presence of multipolar cells and the mechanisms behind their origin.

## Supporting information

**S1 Fig. Percentage of dividing nuclei, cells in cell cycle phase, or cells in mitotic phase do not change with genotype nor rapamycin treatment in patient iPSCs.** (a) Quantification showing average percentage of cells in mitosis per genotype and treatment for experiments included in Fig 1. N = 3 independent replicates per genotype and treatment, [ANOVA]. Error bars = SD. (b) Quantification of relative expression of tuberin in immunoblot shown in d. N = 3 independent replicates, paired students t test. Error bars = SD. (c) Quantification of relative expression of p-S6 Ser240/244 in immunoblot shown in d. N = 3 independent replicates, paired students t test. Error bars = SD. (d) Representative immunoblot showing protein expression of tuberin and p-S6 in patient iPSCs. (e) Quantification showing average percentage of cells in cell cycle phase as determined by DNA content per genotype. N = 3 independent replicates per genotype, [ANOVA]. Error bars = SD. (f) Example images of patient iPSCs showing expression of TUBA (yellow) and Hoechst (teal) to identify DNA to display each cycle of mitosis. (g-j) Quantification showing average percentage of cells in each mitotic phase as determined by mitotic indexing per genotype and treatment. N = 3 independent replicates per genotype, [ANOVA]. Error bars = SD.
(TIF)

**S2 Fig. Percentage of dividing nuclei, cells in cell cycle phase, or cells in mitotic phase do not change with knock out of TSC2 nor rapamycin treatment.** (a) Quantification showing average percentage of cells in mitosis per genotype and treatment for experiments included in Fig 1. N = 3 independent replicates per genotype and treatment, [paired students t test]. Error bars = SD. (b-e) Quantification showing average percentage of cells in each mitotic phase as determined by mitotic indexing per genotype and treatment. N = 2 independent replicates per genotype. Error bars = SD. (f) Quantification showing average percentage of cells in cell cycle phase as determined by DNA content per genotype. N = 2 independent replicates per genotype. Error bars = SD.
(TIF)

**S1 Raw images. Raw images of blots included in these studies.**
(ZIP)

## Acknowledgments

We would like to thank Asa Brockman (Ihrie lab) for the introduction and troubleshooting of CellProfiler workflows, as well as Ihrie, Ess, and Irish lab members for thoughtful discussions of data. We thank Zach Sanchez and Dylan Burnette for assistance with cell imaging. We would like to thank the Vanderbilt Cell Imaging Shared Resource and Nikon Center of Excellence. Microscopy experiments/data analysis/presentation were performed in part through the

use of the Vanderbilt Cell Imaging Shared Resource. Flow Cytometry experiments were performed in the Vanderbilt Flow Cytometry Shared Resource.

## Author Contributions

**Conceptualization:** Mary-Bronwen L. Chalkley.

**Formal analysis:** Mary-Bronwen L. Chalkley, Kevin C. Ess.

**Funding acquisition:** Rebecca A. Ihrie, Kevin C. Ess.

**Investigation:** Mary-Bronwen L. Chalkley, Rachel B. Mersfelder, Maria Sundberg.

**Methodology:** Mary-Bronwen L. Chalkley, Rachel B. Mersfelder, Maria Sundberg, Mustafa Sahin, Rebecca A. Ihrie.

**Project administration:** Kevin C. Ess.

**Resources:** Maria Sundberg, Laura C. Armstrong, Mustafa Sahin.

**Supervision:** Mustafa Sahin, Rebecca A. Ihrie, Kevin C. Ess.

**Writing – original draft:** Mary-Bronwen L. Chalkley.

**Writing – review & editing:** Mary-Bronwen L. Chalkley, Rachel B. Mersfelder, Maria Sundberg, Laura C. Armstrong, Mustafa Sahin, Rebecca A. Ihrie, Kevin C. Ess.

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
