## [Decision Letter · Decision Letter 0]

11 Apr 2023

PONE-D-23-03345Non-Canonical Functions of a Mutant TSC2 Protein in Mitotic DivisionPLOS ONE

Dear Dr. Ess,

Thank you for submitting your manuscript to PLOS ONE. After careful consideration, we feel that it has merit but does not fully meet PLOS ONE’s publication criteria as it currently stands. Therefore, we invite you to submit a revised version of the manuscript that addresses the points raised during the review process.

We have now received comments from two independent referees with significant domain expertise. Both reviewers have found general flaws in the manuscript and its conclusions and have made some criticisms/ questions. Even though one of the referees felt that ths manuscript is too preliminary, I am deciding to give the authors a chance to revise the paper significantly and address the concerns from the reviewers.  Main concerns regarding the need for further proof for the mechanism proposed and the effect of rapamycin  will be necessary to convince the reviewers. Such a significantly revised version could be reconsidered for publication with another round of review.

We look forward to receiving your revised manuscript.

Kind regards,

Manoj B. Menon, PhD

Academic Editor

PLOS ONE

“MS reports grant support from Novartis, Biogen, Astellas, Aeovian, Bridgebio, and Aucta. He has served on Scientific Advisory Boards for Novartis, Roche, Regenxbio, SpringWorks Therapeutics, Jaguar Therapeutics and Alkermes. The other authors have no competing interests to declare.”

Please respond by return email with your amended Competing Interests Statement and we will change the online submission form on your behalf.

“We thank Asa Brockman (Ihrie lab) for introduction and troubleshooting of CellProfiler workflows, as well as members of the Ihrie, Ess, and J. Irish labs for thoughtful discussions of data. We would also like to thank the Vanderbilt Cell Imaging Shared Resource and Nikon Center of Excellence. Experiments/Data analysis/presentation were performed in part through the use of the Vanderbilt Cell Imaging Shared Resource (supported by NIH grants CA68485, DK20593, DK58404, DK59637 and EY08126). This research was supported by awards R01NS118580 (RAI, KCE), R01NS118580-S1 (MBLC), and T32 HD007502 (MBLC).”

“Yes. This research was supported by awards R01NS118580 (RAI, KCE), R01NS118580-S1 (MBLC), and T32 HD007502 (MBLC).  The funders had no role in study design, data collection and analysis, decision to publish, or preparation of the manuscript.”

7. In your Data Availability statement, you have not specified where the minimal data set underlying the results described in your manuscript can be found. PLOS defines a study's minimal data set as the underlying data used to reach the conclusions drawn in the manuscript and any additional data required to replicate the reported study findings in their entirety. All PLOS journals require that the minimal data set be made fully available. For more information about our data policy, please see http://journals.plos.org/plosone/s/data-availability.

Additional Editor Comments:

We have now received comments from two independent referees with significant domain expertise. Both reviewers have found general flaws in the manuscript and its conclusions and have made some criticisms/ questions. Even though one of the referees felt that ths manuscript is too preliminary, I am deciding to give the authors a chance to revise the paper significantly and address the concerns from the reviewers. Such a significantly revised version could be reconsidered for publication with another round of review.

Reviewers' comments:

Reviewer's Responses to Questions

**Comments to the Author**

1. Is the manuscript technically sound, and do the data support the conclusions?

Reviewer #1: Partly

Reviewer #2: No

2. Has the statistical analysis been performed appropriately and rigorously? 

Reviewer #1: Yes

Reviewer #2: Yes

3. Have the authors made all data underlying the findings in their manuscript fully available?

Reviewer #1: Yes

Reviewer #2: Yes

4. Is the manuscript presented in an intelligible fashion and written in standard English?

Reviewer #1: Yes

Reviewer #2: Yes

5. Review Comments to the Author

Reviewer #1: 1. In the results section, the authors say that the expression of the TSC2 in LOF mutatios conditions is decreased compared to WT conditions. Their figure 1A indicates the opposite.

2. It is also not clear with respect to what were the relative expressions of TSC2 calculated in Figure 1B. Authors must elaborate.

3. From Figure 1C, it is hard to understand which cells are undergoing multipolar division in TSC2 +/LOF. It does not appear that there is a cell undergoing multipolar division in this representative image. Authors must provide clear images to substantiate their claim for TSC2 +/LOF.

4. Further, the authors did not indicate the statistical result (Figure 1D and E) for multipolar divisions for TSC2+/LOF vis-à-vis WT and TSC2LOF/LOF. This must be shown to increase confidence in their data.

5. Lastly, in Figure 1F and 2D, they show no significant difference between TSC2 WT and TSC2+/LOF. Considering these, the authors must revisit their conclusion regarding multipolar divisions being aggravated in TSC2 heterozygous condition.

6. Rapamycin is a conventional antagonist of mTORC1. It would be good to see whether rapamycin alters the expression of TSC2 in WT, het or LOF/LOF conditions.

7. Including a test of rapamycin dependent mTORC1 inhibition is recommended.

8. Is lack of TSC2 associated with increased mitotic index?

Reviewer #2: This study proposes a potentially novel dominant function of TSC2/tuberin in regulating mitotic spindle formation directly or indirectly. The novel tuberin function is based on the observation that iPSC derived from a patient with a with a c-terminal 6 amino acid in-frame microdeletion, which is a common mutation found in TSC patients.

1. The observations are interesting, but I found them too preliminary and speculative to support the proposed model for a dominant mitotic function of this mutant TSC2. It is stimulating that loss of TSC2 has been previously shown to be associated with mitotic and cytokinesis defects, pohpshoSer939-TSC2 localized to the corresponding structures. These might be mediated by a TSC2-TACC3 physical interaction (of note TACC3 is not a kinase, as stated: the corresponding discussion sentence & interpretation of previous evidence/references are erroneous).

2. The link with frequent multipolar figures of giant cells observed in hamartomas of TSC patients is also interesting, but it does not provide a plausible explanation beyond the specific mutation assayed in this study. Again, it has been previously described cytokinetic alterations and frequent binucleated cells, potentially linked to overactive microtubule nucleation in TSC2-mutant cells.

3. The proposed dominant effect of the specific TSC2 mutant is based on “negative” assays of KO TSC2 iPSC and siRNA-mediated, but it should be proven through specific in vitro (eg microtubule based) and cellular (using additional models; reconstitution with different mutant forms; specific endogenous gene editing; including variants of the same C-term region, among other potential studies)) assays would be required to finally demonstrate the main message of the study. Indeed, live cell imaging of the mutant vs WT and other TSC2 C-term mutations/variants would be very helpful to decipher and validate the phenotype.

4. The conclusion that the mechanism is independent from rapamycin treatment also appears to be very preliminary as based on a single assay type. Indeed, treating cells for just 24h only includes at best 1 complete cell cycle. The study might also benefit from testing TSC1, constitutive Rheb mutants, among other potential approaches to discern the role of mTORC1 signaling; indeed, phospho-mTOR has also been localized to mitotic structures, but it is not discussed, and associated with mitotic defects.

6. PLOS authors have the option to publish the peer review history of their article (what does this mean?). If published, this will include your full peer review and any attached files.

Reviewer #1: No

Reviewer #2: No

---

## [Author Response · Author response to Decision Letter 0]

3 Sep 2023

Response to Reviewers

We thank the reviewers for the points raised, which we think have strengthened the updated manuscript. Below, our responses are shown in blue for clarity.

Reviewer #1:

1. In the results section, the authors say that the expression of the TSC2 in LOF mutation conditions is decreased compared to WT conditions. Their figure 1A indicates the opposite.

2. It is also not clear with respect to what were the relative expressions of TSC2 calculated in Figure 1B. Authors must elaborate.

With respect to these two questions, the relative expression of TSC2 across genotypes was calculated by normalizing the intensity of the relevant tuberin bands to a loading control (GapDH in Figure 1 and total protein in Supplementary Figure 1). The homozygous TSC2 mutant cells have retained tuberin as the microdeletion form of tuberin is not rapidly degraded. As the GAP domain is impaired in this deletion, this allele is accepted as pathogenic, and in fact is one of the most common pathogenic TSC2 mutations reported in patients with TSC (see ClinVar https://www.ncbi.nlm.nih.gov/clinvar/variation/12402/ and cited references within manuscript). We have better explained this result in the manuscript and more clearly cited the original characterization of these isogenic cell lines, which includes blots consistent with the data shown here.

3. From Figure 1C, it is hard to understand which cells are undergoing multipolar division in TSC2 +/LOF. It does not appear that there is a cell undergoing multipolar division in this representative image. Authors must provide clear images to substantiate their claim for TSC2 +/LOF.

Figure 1 has been revised to make this point more clearly.

4. Further, the authors did not indicate the statistical result (Figure 1D and E) for multipolar divisions for TSC2+/LOF vis-à-vis WT and TSC2LOF/LOF. This must be shown to increase confidence in their data.

 This has now been included. We see an increase in multipolar in the HET but this increase was not statistically significant. We also include additional data (see below) looking at different doses of rapamycin as well as time of exposure.

5. Lastly, in Figure 1F and 2D, they show no significant difference between TSC2 WT and TSC2+/LOF. Considering these, the authors must revisit their conclusion regarding multipolar divisions being aggravated in TSC2 heterozygous condition.

 In the relevant panel (revised Figure 1D), there are not significant differences seen in the number of multi-polar dividing cells in both heterozygote and homozygote mutant cells compared to WT.

6. Rapamycin is a conventional antagonist of mTORC1. It would be good to see whether rapamycin alters the expression of TSC2 in WT, het or LOF/LOF conditions.

 We investigated this extensively (Supplementary Figure 1B) and found no significant differences in tuberin levels with various doses and time of rapamycin exposure.

7. Including a test of rapamycin dependent mTORC1 inhibition is recommended.

 Multiple doses and timepoints of rapamycin treatment are now included - 0.4 nM for 24 and 48 hours as well as 30 nM for 24 hours.

8. Is lack of TSC2 associated with increased mitotic index?

 It is not – this is detailed in Supplementary Figure 1A, as well as cell cycle evaluations shown in Supplementary Figure 1E-J. Supplementary Figure 2 also examines cell cycle parameters in another isogenic paired set of TSC2 WT and KO cells with and without rapamycin.

Reviewer #2: This study proposes a potentially novel dominant function of TSC2/tuberin in regulating mitotic spindle formation directly or indirectly. The novel tuberin function is based on the observation that iPSC derived from a patient with a with a c-terminal 6 amino acid in-frame microdeletion, which is a common mutation found in TSC patients.

1. The observations are interesting, but I found them too preliminary and speculative to support the proposed model for a dominant mitotic function of this mutant TSC2. It is stimulating that loss of TSC2 has been previously shown to be associated with mitotic and cytokinesis defects, pohpshoSer939-TSC2 localized to the corresponding structures. These might be mediated by a TSC2-TACC3 physical interaction (of note TACC3 is not a kinase, as stated: the corresponding discussion sentence & interpretation of previous evidence/references are erroneous).

 A TACC3 discussion has been incorporated into the manuscript. We thank the reviewer for pointing this out. We also think our revised data makes a more compelling argument for this potential mechanism of this mutant TSC2 allele, but have been careful to not overstate/overinterpret our findings.

2. The link with frequent multipolar figures of giant cells observed in hamartomas of TSC patients is also interesting, but it does not provide a plausible explanation beyond the specific mutation assayed in this study. Again, it has been previously described cytokinetic alterations and frequent binucleated cells, potentially linked to overactive microtubule nucleation in TSC2-mutant cells.

 We think the strength of our study is the rigorous use of different TSC2 mutant iPSC to investigate cell cycle and multipolar cells – prior work most often examined these features in lines that were not derived from patients. The reviewer is correct that this manuscript focuses on the specific 6 amino acid microdeletion found in these lines; however, we think this work remains impactful for two reasons: first, this mutation is among the most common pathogenic mutations found in patients with TSC, and second, many studies of TSC models rely on full knockouts that were generated from wild-type human cells. The work detailed here contributes to the increasing understanding that comparison of multiple disease-associated genotypes will be important to finding common and idiosyncratic effects of the mutations found in the clinic.

3. The proposed dominant effect of the specific TSC2 mutant is based on “negative” assays of KO TSC2 iPSC and siRNA-mediated, but it should be proven through specific in vitro (eg microtubule based) and cellular (using additional models; reconstitution with different mutant forms; specific endogenous gene editing; including variants of the same C-term region, among other potential studies)) assays would be required to finally demonstrate the main message of the study. Indeed, live cell imaging of the mutant vs WT and other TSC2 C-term mutations/variants would be very helpful to decipher and validate the phenotype.

 We agree further assays would provide additional evidence but think our conclusions are sound based on the available evidence. We did transfect TSC2 homozygous mutant cells with WT and mutant (6 amino acid deletion) tuberin expression constructs that had a fluorescent tag to enable live cell imaging. While fluorescence corresponding to mutant tuberin was clearly seen at the beginning of the experiment, cells expressing only mutant tuberin were not detectable several days later. This would suggest that the non-physiological overexpression of mutant tuberin led to cell death and supports a dominant negative mechanism. However, this is another “negative” assay and we decided to not include these data in our manuscript.

4. The conclusion that the mechanism is independent from rapamycin treatment also appears to be very preliminary as based on a single assay type. Indeed, treating cells for just 24h only includes at best 1 complete cell cycle. The study might also benefit from testing TSC1, constitutive Rheb mutants, among other potential approaches to discern the role of mTORC1 signaling; indeed, phospho-mTOR has also been localized to mitotic structures, but it is not discussed, and associated with mitotic defects.

 In response to this reviewer concern, we have expanded the rapamycin treatment data using 0.4 nM at 24 and 48 hours as well as 30 nM at 24 hours (Figure 1D and Supplementary Figure 1. The other suggested experiments (looking at TSC1, Rheb constitutive expression, phospho-mTOR) are beyond the scope of this report but are excellent suggestions for future work.

---

## [Decision Letter · Decision Letter 1]

12 Sep 2023

Non-Canonical Functions of a Mutant TSC2 Protein in Mitotic Division

PONE-D-23-03345R1

Dear Dr. Ess,

We’re pleased to inform you that your manuscript has been judged scientifically suitable for publication and will be formally accepted for publication once it meets all outstanding technical requirements.

Kind regards,

Manoj B. Menon, PhD

Academic Editor

PLOS ONE

Additional Editor Comments (optional):

The manuscript has been significantly revised and all constructive criticisms seems to have been addressed. Both the referees are now happy to recommend acceptance of the revised manuscript.  Looking forward to seeing the publication online soon. Congrats to the team.

Reviewers' comments:

Reviewer's Responses to Questions

**Comments to the Author**

1. If the authors have adequately addressed your comments raised in a previous round of review and you feel that this manuscript is now acceptable for publication, you may indicate that here to bypass the “Comments to the Author” section, enter your conflict of interest statement in the “Confidential to Editor” section, and submit your "Accept" recommendation.

Reviewer #1: All comments have been addressed

Reviewer #2: All comments have been addressed

2. Is the manuscript technically sound, and do the data support the conclusions?

Reviewer #1: Yes

Reviewer #2: Yes

3. Has the statistical analysis been performed appropriately and rigorously? 

Reviewer #1: Yes

Reviewer #2: Yes

4. Have the authors made all data underlying the findings in their manuscript fully available?

Reviewer #1: Yes

Reviewer #2: Yes

5. Is the manuscript presented in an intelligible fashion and written in standard English?

Reviewer #1: Yes

Reviewer #2: Yes

6. Review Comments to the Author

Reviewer #1: (No Response)

Reviewer #2: The authors have responded adequately to the major initial comments. The authors have left mechanistic questions for future studies, which I do agree as the described observation is relevant per se.

7. PLOS authors have the option to publish the peer review history of their article (what does this mean?). If published, this will include your full peer review and any attached files.

Reviewer #1: **Yes: **Anita Roy

Reviewer #2: No

---

## [Editor Report · Acceptance letter]

25 Sep 2023

PONE-D-23-03345R1 

Non-Canonical Functions of a Mutant TSC2 Protein in Mitotic Division 

Dear Dr. Ess:

I'm pleased to inform you that your manuscript has been deemed suitable for publication in PLOS ONE. Congratulations! Your manuscript is now with our production department. 

Kind regards, 

on behalf of

Dr. Manoj B. Menon 

Academic Editor

PLOS ONE